# A New Lignan from *Annona squamosa* L. (Annonaceae) Demonstrates Vasorelaxant Effects In Vitro

**DOI:** 10.3390/molecules28114256

**Published:** 2023-05-23

**Authors:** Camilo Di Giulio, Juan Manuel Gonzalez Guzman, Joao Victor Dutra Gomes, Young Hae Choi, Pérola O. Magalhães, Yris M. Fonseca-Bazzo, Damaris Silveira, Omar Estrada

**Affiliations:** 1Centre of Biophysics and Biochemistry, Venezuelan Institute for Scientific Research, Caracas 1020A, Venezuela; digiulic@gmail.com (C.D.G.); jmgguzman@gmail.com (J.M.G.G.); 2Faculty of Health Sciences, University of Brasilia, Darcy Ribeiro University Campus, Asa Norte, Federal District, Brasília CEP 70910-900, Brazil; dutra.joaovictor@gmail.com (J.V.D.G.); perolam@hotmail.com (P.O.M.); yrisfonseca@hotmail.com (Y.M.F.-B.);; 3Natural Products Laboratory, Institute of Biology, Leiden University, 2333 BE Leiden, The Netherlands; y.h.choi@biology.leidenuniv.nl

**Keywords:** furofuran lignan, vasorelaxant, *Annona squamosa*

## Abstract

Esquamosan, a new furofuran lignan, has been isolated by bio-guided assays from the methanolic extract of the leaves of *Annona squamosa* L., and its structure was elucidated by spectroscopic methods. Esquamosan inhibited the rat aortic ring contraction evoked by phenylephrine in a concentration-dependent manner and showed an inhibitory effect on vasocontraction of the depolarized aorta with high-concentration potassium. The vasorelaxant effect by esquamosan could be attributed mainly to the inhibition of calcium influx from extracellular space through voltage-dependent calcium channels or receptor-operated Ca^2+^ channels and also partly mediated through the increased release of NO from endothelial cells. The ability of esquamosan to modify the vascular reactivity of rat aortic rings incubated with high glucose (D-glucose 55 mM) was then evaluated, and this furofuran lignan reverted the endothelium-dependent impairment effect of high glucose in rat aortic rings. The antioxidant capacity of esquamosan was assessed using DPPH and FRAP assays. Esquamosan exhibited a similar antioxidant capacity compared to ascorbic acid, which was used as a positive control. In conclusion, this lignan showed a vasorelaxant effect, free radical scavenging capacity, and potential reductive power, suggesting its potential beneficial use to treat complex cardiometabolic diseases due to free radical-mediated diseases and its calcium antagonist effect.

## 1. Introduction

With nearly 20 million deaths per year, cardiovascular disease (CVD) represents one-third of global mortality [1]. The most common manifestations that lead to these deaths of cardiovascular origin are ischemia, arrhythmias, and arterial hypertension, problems also associated with high morbidity and high costs to health systems [2]. Arterial hypertension generates particular interest because of its high prevalence, estimated at 30% in the adult population worldwide, and the low success in its pharmacological treatment, which will continue to be a risk factor with a high impact on cardiovascular morbidity and mortality in the coming decades. [3,4]. Although we have therapeutic tools with cardiovascular action, there are problems such as its pharmacological tolerance, its high levels of toxicity, especially by antiarrhythmic drugs, low efficacy in diseases such as hypertension [5], and the risks of polymedication, which has led to the active search for new molecules that shape multiple cardiac functions [6]. The search for new bioactive molecules in natural products is an interesting alternative due to their safety profiles and their expanded use as traditional medicine [7,8]. Evidence suggests that molecules such as cardenolides, bufadienolides, flavonoids, phenols and saponins, among others, are cardiomodulators and could be the key to finding and developing new drugs [9]. *Annona squamosa* L. is a tree that has multiple pharmacological applications in the traditional medicine of India, including its popular use for the treatment of cardiometabolic disease, which has been validated in experimental models [10]. The vasorelaxant effect of the extract could be partially attributed to kaurenoic acid and cyclopeptides [11,12]. Given that it is convenient to experimentally validate the entire active ingredients and that a direct effect on smooth muscle contraction could explain its effect on blood pressure, we carried out a bio-guided study on the ethanolic extract of *Annona squamosa* leaves in order to identify the main molecule responsible for the vasorelaxant effects on rat aortic rings. Kumar et al. (2021) described several biological activities of extracts from *Annona squamosa* leaves, including anticancer, antidiabetic, antiobesity, lipid-lowering, and hepatoprotective functions [13]. As far as we know, this is the first bio-guided report of the cardioprotective effects of *Annona squamosa* leaves, which leads to an isolated active compound.

## 2. Results

### 2.1. Bio-Guided Study of Methanolic Extract of Annona Squamosa

Fresh leaves of *Annona squamosa* (320 g) were extracted with 2 L of methanol, yielding 46.0 g of methanolic extract, after evaporation in a vacuum using a rotary evaporator. This extract was named by us as ME. The isolation of the active compound from ME was initiated by its fractionation based on differences in the solubility of ME at room temperature, see Figure 1. A mixture of methanol: water in equal proportion (300 mL × 3) was added to ME, and we obtained two fractions, a methanol-water soluble fraction named MWSF (26.6 g) and a methanol-water insoluble fraction named MWIF (19.2 g); after evaporation of the solvent of both fractions by a vacuum, only MWIF retained the main vasorelaxant effect on rat aortic rings assay. A portion of MWIF (2 g) was chromatographed on Sephadex LH20 using ethanol as an eluent to yield three subfractions named MWIF(I-III). Subfraction MWIF-III retained the main vasorelaxant effect, from which compound **1** (410 mg) was separated by column chromatography (CC) on RP-18, eluted with a mixture of acetone/water (3:2). Subfractions MWIF-II and MWIF-I showed little or no vasorelaxant effect, respectively; from subfraction MWIF-II (400 mg), five known kauren diterpens were isolated by CC on RP18 using a mixture of ethanol-THF-water (75:5:20) as eluent: ent-kaur-16-en-19-ol (2) (118 mg), kaurenoic acid (3) (15 mg), 17-hydroxy-16β-ent-kauran-19-al (4) (87 mg), 17-hydroxy-ent-kaur-15-en-19-al (5) (90 mg), and annosquamosin C (6) (71 mg); these diterpenoids were identified by comparing its NMR spectroscopic data with those in the literature [14]. In order to identify other compounds in ME, a portion of MWSF (1 g) was repeatedly extracted with acetone obtaining two new subfractions, a brownish residue named acetone insoluble fraction (AIF) and a yellowish solution that was evaporated in vacuo from which a yellowish residue named AF was yielded (11.7 g). A portion of AF (1 g) was fractionated on Sephadex LH-20 CC using methanol as eluent to give three subfractions named AF(I-III). Catechin (7) (200 mg) was crystallized from subfraction AF-I in water. From subfraction, AF-III kaempferol (8) (42 mg) was separated by CC on RP-18, eluted with a mixture of acetone/water (3:2). From subfraction AF-II kaempferol-3-*O*-rhamnoside (9), quercetin -3-*O*-rhamnoside (10) and quercetin 3,4′-*O*-diglucoside (11) were isolated by CC on RP18 using a mixture of acetonitrile-THF-water (55:5:40) as eluent; these flavonoids were previously isolated by Zhu et al. [15] and were identified by comparison of its NMR spectroscopic data with those reported in the literature [16]. The separation described above is represented in Figure 1.

### 2.2. Structural Elucidation of Compound ***1***

Compound **1** was isolated as a colourless oil with optical rotation ${\left[\alpha \right]}_{D}^{25}$ = + 215.22 (c 1.2, CHCl_3_), and its molecular formula was deduced to be C_21_H_24_O_5_ by positive HR-ESI-QTOF-MS, exhibiting an ion at *m*/*z* 379.1678 [M + Na]^+^. The ^1^H NMR spectrum of compound **1** displayed resonances for two oxymethylene groups at δH 3.87 (H-9eq) and 4.21 (1H, dd, *J* = 9.2, 6.1 Hz, H-9ax) and at δH 4.24 (1H, dd, *J* = 9.2 and 6.1 Hz, H-9′eq) and 3.87 (H-9′ax), two oxymethine groups at δH 4.72 (1H, d, *J* = 4.1 Hz, H-7) and 4.75 (1H, d, *J* = 4.1 Hz, H-7′), and two methine groups at δH 3.09 (2H, m, H-8 and H-8′), indicating the presence of a furofuran moiety. The ^1^H NMR spectrum also showed resonances for three methoxy groups at δH 3.75 (3H, s), 3.84 (3H, s), and 3.87 (3H) and seven aromatic protons as one AA’MM’ system at δH 7.26 (2H, 8.7 Hz) and 6.87 (2H, 8.7 Hz), representing the presence of two p-disubstituted benzene rings, two singlets at δH 6.91, and one overlapped multiplet at 6.83 ppm with the representation of the presence of 3,4-disubstituted benzene rings. The ^13^C and DEPT NMR spectroscopic data showed twenty-one signals (see in Appendix A). Except for the resonances for tree methoxy groups, the remaining eighteen carbon signals were consistent with a furofuran lignan derivative similar to those of membrine and epimembrine [17]. Although the bulk of the structure of 1 is similar to these known furofuran lignans, the variance in their NMR data (Table 1) indicates that the difference among them should correspond to the relative orientation of the aromatic groups. The ring junction of the bistetrahydrofuran ring system was established as axial conformation based on the couple constant of H-7/H-8 and H-7′/H-8′(4,1 Hz) [18] and NOESY spectra which exhibited correlation among signals at 3.09 (H-8 and H-8′), 3.87(H-9eq), 4.24(H-9′eq), and 4.75–4.72 ppm (H-7 and H-7′), indicating axial orientations of both aromatic rings. Finally, the specific rotation of compound **1** is +215.22°; according to Greger and Hofer [19], all (+)-sesamin type lignans belong to the same series with the absolute configuration R at the bridge carbons 8 and 8′. For this reason, the configuration of compound **1** should be R at carbons 8 and 8′. Due to the fact that aromatics substituents of compound **1** are diaxial and showed specific dextrorotatory rotation, it confirms the R configuration at carbons 7 and 7′; all (+)-sesamin type lignans with absolute configuration R at bridge carbons 8 and 8′ have diaxal aromatic substitution belong to the same series with the absolute configuration R at the benzylic carbon 7 and 7′ [19].



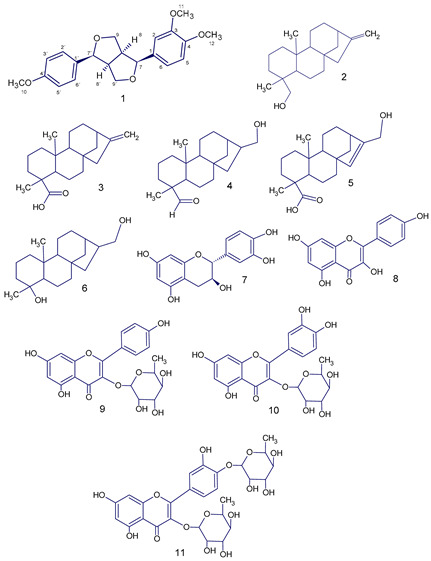



**Table 1 molecules-28-04256-t001:** ^13^C (150 MHz) and ^1^H (600 MHz) NMR data of compound **1** in CDCl3. δ in ppm, *J* in Hz.

Position	δ_H_ (*J* _Hz_)	δ_C_
8	3.09 m	55.0
7′	4.75 d (4.1)	86.5
9ax	4.21 dd (9.2, 6.1)	72.5
9eq	3.87 overlapped	72.5
8′	3.09 m	55.0
7	4.72 d (4.1)	86.5
9′ax	3.87 overlapped	72.5
9′eq	4.24 dd (9.2, 6.1)	72.5
1	-	134.7
2	6.83 m	109.2
3	-	149.6
4	-	150.2
5	6.91 2 s	112.5
6	6.83 m	118.2
1′	-	135.3
2′	6.87 d (8.7)	114.5
3′	7.26 d (8.7)	128.5
4′	-	160.1
5′	7.26 d (8.7)	128.5
6′	6.87 d (8.7)	114.5
10-OCH_3_	3.75	
11-OCH_3_	3.84	
12-OCH_3_	3.87	

### 2.3. Vasorelaxant Effect of Esquamosan

In the first series of experiments, the effect of esquamosan on submaximal PE and KCl-induced contraction of aortic rings (E+) and (E−) were examined in the following manner: after the equilibration period, arteries were precontracted with PE (0.1 µM) or KCl 50 mM, and when a stable level of tone was established, cumulative concentration–response curves to esquamosan were constructed (1 nM to 100 µM). In time-match-control experiments, we determined if the contractions induced by PE (0.1 µM) and KCl (50 mM) were stable enough for the period required to construct the concentration–relaxation curve. Rings similarly precontracted with PE or KCl and treated with vehicle (dimethylsulfoxide, DMSO 0.5%) allowed us to verify that it did not have an effect on the precontractile force. All subsequent series of experiments were carried out in denuded arteries. In the second set of experiments, esquamosan (1 µM) was added to the organ bath 10 min before constructing the concentration–response curve for PE (1 pM to 10 µM) or KCl (5 mM to 80 mM). Tension was expressed as the percentage of KCl or PE-induced contraction. As showed in Figure 1A,B, esquamosan induced a concentration-dependent relaxation in aortic rings (E+) or precontracted (E−) with PE (IC50 = 1.05 ± 0.08 µM, Emax = 92.4 ± 3.5%) and KCl (IC50 = 1.45 ± 0.05 µM, Emax = 90.4 ± 6.5%). The baseline tension of rat aortic rings was not modified by esquamosan tested up to 100 µM. In addition, esquamosan (1 µM) depressed almost half of the maximum response of PE or KCl concentration–effect curves (*p* < 0.05), exhibiting a non-competitive antagonism (Figure 1C,D).

**Figure 1 molecules-28-04256-f001:**
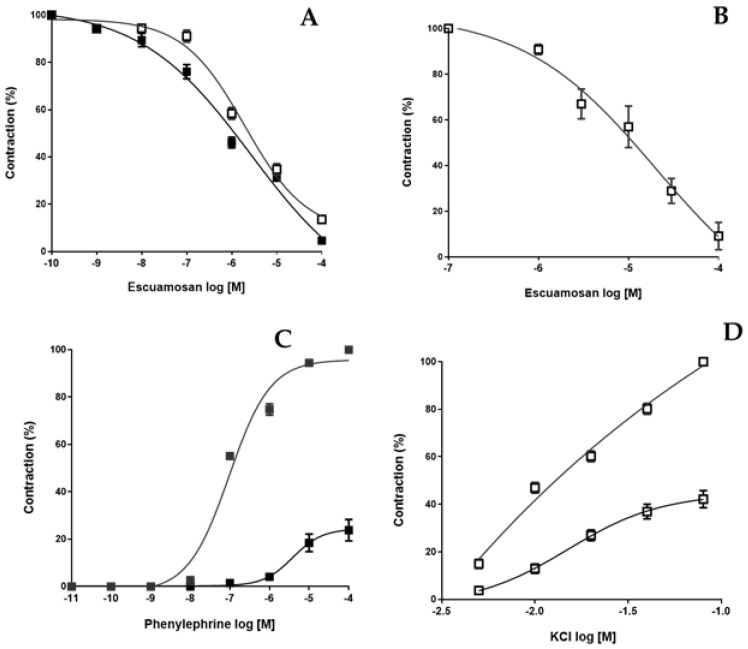
Vasorelaxant effect induced by esquamosan on isolated rat aortic rings. Cumulative concentration-response curves to esquamosan (1 nM to 100 µM) were constructed in panel (**A**), PE (0.1 µM)-precontracted aortic rings (E+, ■) and (E−, □) or in panel (**B**), KCl (50 mM)-precontracted aortic rings (E−). In panel (**C**), PE and panel (**D**) KCl cumulative concentration-effect curve was constructed in the absence or in the presence of esquamosan (1 µM), respectively. Results are expressed as the mean ± SD of six independent experiments. Unpaired *t*-test with Welch’s correction.

### 2.4. Contribution of Endothelium-Derived Relaxing Factors on the Vasorelaxant Effect of Esquamosan

The contribution of nitric oxide (NO) and cyclooxygenase-derived products on the vascular response elicited by esquamosan was examined by pretreatment of aortic rings with the NO synthase inhibitor L-NAME (100 µM) and the cyclooxygenase inhibitor indomethacin (10 µM), respectively, which were added to the organ bath 10 min before contracting the arterial rings (E+) with PE (0.1 µM). The vasorelaxant effect of esquamosan was unaffected by the pretreatment of PE-precontracted aortic rings (E+) with L-NAME or indomethacin.

### 2.5. Evaluation of the Esquamosan Effect on CaCl_2_, (S)-(−)-BAY-K-8644: And Caffeine-Induced Precontractions

As can be seen in Figure 2A, pretreatment with esquamosan attenuated the CaCl_2_-induced contraction of the denuded rat aorta exposed to a Ca^2+^-free medium that contains KCl. CaCl_2_ induced a concentration-dependent contraction of rat aortic rings. Pre-incubation of the rat aortic rings with esquamosan at 1, 5, or 10 µM significantly decreased in a noncompetitive manner the contraction evoked by CaCl_2_. In order to determine if the inhibition of the L-type calcium channel could contribute to the vasorelaxant effect of esquamosan, aorta ring preparations were pre-incubated with a depolarizing solution of KCl 20 mM for 10 min, a procedure performed to obtain a better response for the contractile agent. Then, after 10 min of incubation in the presence of esquamosan (1, 5, and 10 µM), the agonist by (S)-(−)-BAY-K-8644 (0.01–1 μM), an L-type Ca^2+^ channel activator, was added to induce a sustained contraction. Contractions induced by (S)-(−)-BAY-K-8644 were reduced in a dose-dependent manner by esquamosan, see Figure 2B. On the other hand, in order to investigate whether esquamosan could interfere with the Ca^2+^ release from intracellular stores, the inhibitory effects of esquamosan on caffeine-induced contractions in the absence of extracellular Ca^2+^ were determined in endothelium-denuded aortic rings, see Figure 2C. A normal Krebs solution was replaced with a Ca^2+^-free solution containing EGTA (1 mM) for 15 min and was then washed with a Ca^2+^-free solution. The rings were stimulated with caffeine (20 mM). The contraction induced by caffeine was obtained after 10 min of incubation in the presence of esquamosan (0.1 µM to 100 µM). Esquamosan induced a calcium-released blockade from intracellular stores since the transient contractions induced by caffeine, an activator of the ryanodine receptor, were tackled in a dose-dependent manner.

### 2.6. Esquamosan Reverted the Endothelium-Dependent Impairment Effect of High Glucose in Rat Aortic Rings

The fourth experiment was designed to evaluate the effect of esquamosan on the impairment of endothelium-dependent relaxation induced by high glucose levels. As shown in Figure 3, when aortic rings were pre-incubated for 6 h with HG-Krebs (closed black triangle), the Emax of ACh decreased from 75 ± 5% (*n* = 5) to 17 ± 2% (*n* = 6). However, the aortic rings incubated with esquamosan (1 µM) (closed black squares) improved the capability of relaxation induced by ACh in HG-Krebs solution, indicating that the deleterious effects of HG levels were noticeably attenuated. Esquamosan mildly improved the relaxation ability of the control rings, see Figure 3.

### 2.7. Antioxidant Capacity of Esquamosan

Esquamosan was tested for antioxidant capacity using a DPPH scavenging assay and TPTZ-complex-reducing (FRAP) activity. Our results indicated that esquamosan had favourable antioxidant capacity, as expected for lignan compounds, scavenging activity, and reduced power ability. However, it was less than that of ascorbic acid (positive control), as in Figure 4. EC_50_ values in the DPPH assay were 56.8 ± 1.3 and 5.39 ± 0.1 μg/mL (159.4 and 30.6 μM) for esquamosan and ascorbic acid, respectively. The FRAP activity results were 32.6 ± 6.5 and 8.9 ± 0.3 μg/mL, equivalents to 100 µM of Fe^2+^ for esquamosan and ascorbic acid, respectively.

## 3. Discussion

*Annona squamosa* is a plant that has been used for the treatment of diabetes and hypertension [20]. Previous studies have reported the isolation from this plant of vasodilators such as cyclosquamosin B and kaurenoic acid, which partially explain the vascular effects of the extract from the leaves of this plant. However, none of the former compounds has been reported with potential cardiometabolic effects. For this reason, our research group decided to carry out a bio-guided study in rat aortic rings assay. In this research, we isolated a new furofuran lignan by bio-guided assays, which is named by us as esquamosan, which demonstrated a vasorelaxing effect mainly attributed to the inhibition of calcium influx from the extracellular space. Among other reasons, diabetic vascular complications are caused by prolonged exposure to high glucose levels, and as a consequence, oxidative stress is considerably increased because of the production of reactive oxygen species (ROS) in the vascular wall; this pathological condition impairs endothelial function and is thought to mediate cardiovascular disease [21,22]. Thus, we considered the evaluation of the ability of esquamosan to modify the vascular reactivity of rat aortic rings incubated with high glucose (55 mM D-glucose); this lignan reversed the endothelium-dependent impairment effect of high glucose on rat aortic rings. Most of the mechanisms reported in the literature underlying hyperglycemia-induced diabetic vascular damage focus on five main mechanisms: the polyol pathway, increased intracellular formation of advanced glycation end-products, increased expression of the receptor for advanced glycation end-products and their activating ligands and activation of protein kinase C isoforms. The mentioned mechanisms are activated by the mitochondrial overproduction of ROS. Thus, overall vascular function is dependent upon the balance of oxidant and antioxidant mechanisms, which determines endothelial function. In this sense, we evaluated the antioxidant effect of esquamosan through the DPPH and FRAP assays.

Lignans are secondary metabolites with a wide range of biological activities, such as antimicrobial, anti-inflammatory, anti-neurodegenerative, antioxidant, vasorelaxant, antitumor, and antiviral [23]. Kose et al. (2021) [24] also investigated the antioxidant potential of lignans. As in our present study, DPPH and iron reduction assays were used, and other assays, such as the ABTS radical, were additionally evaluated. The results of antioxidant activities ranged from 10 to 1000 μg/mL approximately, depending on the lignan evaluated. It was discussed about the influence of groups linked to the core structure, where hydroxyl groups contribute to antioxidant activity more strongly than methoxyl groups, as in the case of esquamosan. Forsythoside L and its aglycon Forsythol L [25] were isolated from the aerial parts of *Forsythia suspensa* (Thunb.) Vahl and evaluated in the DPPH radical scavenging experiment and ferric-reducing ability of plasma (FRAP) experiment. These compounds exhibited antioxidant activity with IC_50_ values ranging from 112.49 to 153.58 μM in the DPPH experiment, which was similar to our result for esquamosan (159.4 μM) in the same experiment.

Lignans such as eudesmin, saucerneol, saucerneol D, saucerneol E, manassantin A, cudebin, and gomision J have shown a vasorelaxing effect dependent on vascular endothelium in experiments performed on rat aorta rings [26,27,28]. Only Phyllanthin, Hypophyllanthin [29], yangambin, and now esquamosan have demonstrated a vasorelaxing effect independent of the vascular endothelium and mediated by blocking the movement of calcium within the smooth muscle. It would be interesting for future molecular modelling studies to find out which are the common structural characteristics that give these compounds such vasorelaxing properties.

## 4. Materials and Methods

### 4.1. General Information

RediSep^®^ Rf Reversed-phase C18 was used for low-pressure CC and silica gel 60 RP18 F254 (E. Merck, Darmstadt, Germany) for TLC on glass (Merck, Rahway, NJ, USA). Dimethylsulfoxide (DMSO), adenosine 5′-diphosphate sodium salt (ADP), and Sephadex^®^ LH-20 were obtained from Sigma Aldrich (St. Louis, MO, USA). 1D (^1^H, ^13^C) and 2D NMR (HSQC, HMBC, COSY and NOESY) spectra were obtained using a Bruker Ascend 600 (600 MHz for 1H and 150 MHz for ^13^C) in deuterated chloroform (CDCl_3_; Sigma-Aldrich). All solvents used were of HPLC grade quality and obtained commercially from Sigma-Aldrich. The following drugs used for vascular reactivity experiments were also purchased from Sigma-Aldrich: D-glucose, KCl, CaCl_2_, (S)-(−)-BAY-K-8644, ACh, and PE. Reversed-phase C18 was used for low-pressure column chromatography and silica gel 60 RP18 F254 (Merck) for TLC on glass (Merck). TLC plates were sprayed with a saturated solution of ceric sulfate in 65% sulfuric acid (Sigma-Aldrich) and heated to 120 °C.

### 4.2. Experimental Procedures with Animals

The animal studies were evaluated and approved by the Bioethics Commission for Investigations in Animals at the Instituto Venezolano de Investigaciones Cientificas (IVIC), protocol 201417, approved on November 2017, in accordance with the Code on Bioethics and Biosecurity (2008) established by the Bioethics Commission National Fund on Science and Technology (FONACIT) under the national legislation. Male Sprague-Dawley rats weighing 240–350 g were used for vascular reactivity assays and were obtained from the animal care service of IVIC. They were housed at room temperature (21 ± 2 °C) and light–dark (06:00–18:00 h), with a maximum of four rats per acrylic, transparent, rectangular cage, with wood-derived bedding. Food and tap water were freely available at the conventional animal facilities of the IVIC.

### 4.3. Plant Material

*Anonna squamosa* leaves were collected in September 2018 at the National Park Henri Pittier (Edo Aragua, Venezuela). The species was identified by Dr Alfonso Cardozo (Department of Botany, Venezuelan Central University). A voucher specimen was deposited in the herbarium at the Agronomy Faculty of the Venezuelan Central University (voucher number: AC21992).

### 4.4. Spectral Data

(+)-Esquamosan (1) Yellowish oil [α]*_D_* + 215.22° (CHCl3; c 1.2) HRMS: *m*/*z*: 379.1678 ([M + Na]+ (calcd for C_21_H_24_NaO_5_, 379,1621) MS *m*/*z* (rel int.) 395.1445(10), 379.1678 ([M+ Na])+ (100), 355.1726 (15), 339, 1772 (25),198.1006 (8), 121.1138 (10).IR ν max cm^−1^ 1591, 1515, 1464, 1417, 1375, 1071 and 1031.1

### 4.5. Vascular Reactivity Azssays

The rats were anaesthetised with sodium pentobarbital (40 mg/kg body weight, intraperitoneal) and the chests were opened by midline incision to isolate the descending thoracic aorta. After removal of the superficial fat and connective tissue, the rings with intact endothelium (E+) were placed in a 10 mL organ chamber containing a physiological salt solution, constantly bubbled with a mixture of 95% O_2_/5% CO_2_ and maintained at 37 °C. The composition of the physiological salt solution was as follows (mM): NaCl, 118; KCl, 4.7; KH_2_PO_4_, 1.2; MgSO_4_, 1.2; NaHCO_3_, 15; glucose, 5.5; and CaCl_2_, 2.5. The rings were stretched until an optimal basal tension of 2 g was reached, previously determined by length–tension relationship experiments, and then were allowed to equilibrate for 2 h with the bath fluid being changed every 15–20 min. The mechanical stability of the system was tested by adding KCl 50 mM three times. Aortic rings were repeatedly washed and allowed to re-equilibrate for an additional 30 min. Endothelial integrity was assessed qualitatively by the degree of relaxation caused by ACh (10 μM) in the presence of the contractile tone induced by PE (1 μM). In some rings (E−), the endothelial layer was removed immediately after dissection by gently rubbing the luminal surface with a small wooden stick. Only one experiment was carried out in each aorta ring. Isometric tensions were recorded on a polygraph (PowerLab 4/26; PanLab) by means of a force-displacement transducer ML201. Data were fed to a computer, and a LabChart system (PanLab) was used to convert acquired data into the digital form. For esquamosan assays, PE 1 μM precontracted E+ and E− rings were exposed to increasing (0.01–100 μM) esquamosan concentrations. For HG experiments, the first ring (control) was incubated in KHB, the second ring was incubated in KHB with high glucose (HG; 55 mM), and the third ring was incubated in KHB with HG plus esquamosan 1 μM. A KHB plus esquamosan 1 μM group was also recorded. All rings were incubated for 6 h. After this period, rings were precontracted with PE 1 μM, and once a stable contraction was achieved, cumulative concentration–response curves to ACh were obtained. Maximum relaxation (Emax) and concentration producing the half-maximal effect (EC50) were determined from each concentration–response curve. The relaxation from the precontracted level to the baseline was considered 100% relaxation.

### 4.6. Determination of Antioxidant Capacity of Ferric-Reducing Antioxidant Power (FRAP) Assay

The antioxidant capacity of esquamosan was investigated by the FRAP assay performed as described by Kelman et al., with few modifications [30]. The FRAP working solution was made by mixing 300 mM acetate buffer (pH 3.6), 10 mM 2,4,6-tripyridyl-S-triazine (TPTZ) solution in 40 mM HCl, and 20 mM FeCl_3_•6H_2_O in a 10:1:1 ratio and heated to 37 °C in a water bath prior to use. Then, 150 µL of FRAP working solution was added to 20 µL of the sample. After 8 min incubation at room temperature and protection from light, the absorbance was measured at 595 nm using a multimode plate reader (EnSpire, Perkin-Elmer, Singapore). The absorbance value of each sample containing FRAP working solution but without TPTZ was used as a background control. In every independent experiment, a standard curve was made using known Fe^2+^ concentrations between 10 and 350 μM of FeSO_4_•7H_2_O, and an equation curve was calculated. The results were expressed as reduced iron equivalents (Fe^2+^, µM). All samples were measured in duplicate, and the presented results are the mean of at least three independent experiments.

### 4.7. Determination of Antioxidant Capacity of 2,2-Diphenyl-1-picrylhydrazyl (DPPH) Assay

The antioxidant capacity was investigated by the scavenging activity of the radical DPPH, based on Les et al., with some modifications [31]. In total, 100 µL of a DPPH ethanolic solution (0.2 mM) was added to 100 µL of different concentrations of esquamosan diluted in ethanol. All reactions contained 2.5% water in the end. After 30 min of reaction at room temperature and protection from the light, absorbance was measured at 515 nm using a multimode plate reader (EnSpire, Perkin-Elmer, Singapore). As a negative control, 100 µL DPPH solution added to 100 µL of ethanol was used. The sample background absorbance was subtracted before calculating the DPPH scavenging activity. Scavenging activity (%) = [(Abs control − Abs sample)/Abs control] × 100. The results were expressed as the relative concentration required to reduce the DPPH concentration by 50% (EC50). Ascorbic acid was used as a positive control of antioxidant activity in both assays.

### 4.8. Statistics

Values are expressed as means and standard deviation (SD). Statistical analysis was performed by applying one-way ANOVA using GraphPad Prism (version 6.1) software. Statistical comparisons between two data means were made using an unpaired *t*-test with Welch’s correction, and *p*-values less than 0.05 were considered statistically significant. The vasorelaxing response was expressed in terms of the percentage of decrease in the maximal contraction caused by PE (1 μM). The IC_50_ value was defined as the concentration of the compound that reduced the maximum contraction elicited by PE by 50% and was calculated from a concentration–response curve, which was analysed by nonlinear regression (curve fit) using GraphPad Prism (Version 6.1). Antioxidant analyses were performed in triplicate, and data are expressed as the mean ± standard deviation.

## 5. Conclusions

Using bioguided studies in rat aortic ring assays we isolated a novel ligand named esquamosan which possesses endothelium-independent vasorelaxant properties in isolated rat aorta. This vasorelaxant effects is mediated by inhibiting both Ca^2+^ influx through voltage-dependent calcium channels and Ca^2+^ release from intracellular stores channels in vascular smooth muscle cells. Esquamosan also demonstrates a good scavenging effect and reducing power that could explain its ability to tackle the loss of vascular endothelial reactivity at high glucose concentrations in aortic ring assay. The direct vasorelaxant effect and the vasoprotective properties of esquamosan in the context of oxidative stress demonstrated in vitro suggests its potential beneficial use in the treatment of complex cardiometabolic diseases.

## Data Availability

The original data presented in the study are included in the article; further inquiries can be directed to the corresponding authors.

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
