# Peer review of "A New Lignan from Annona squamosa L. (Annonaceae) Demonstrates Vasorelaxant Effects In Vitro"

_molecules, 2023, doi:10.3390/molecules28114256_

Round 1

Reviewer 1 Report

molecules-2386105

A New Lignan from Annona squamosa L. (Annonaceae) demonstrate vasorelaxant effects in vitro

Camilo Di Giulio et al., 2023

Topic Antioxidant Activity of Natural Products

The information included for structure 1 could be improved and supported with additional experiments considering the availability of powerful NMR equipment.

1-    Please, assign a number to Figure 1. Structure of squamosan. , which will allow a better reading of Table 1.

2-    Please, assign a number to the carbons in Figure 1, for a better reading of the 13C data that you mention in the section 4.4. Spectral Data 13 C.

3-    Section  4.1. General Information

“1H and 13C NMR spectra were obtained using a Bruker DRX 500

(500 MHz for 1H and 125 MHz for 13C) in deuterated dimethylsulfoxide (DMSO-d6; Sigma-Aldrich).”

Considering that the authors have a robust NMR team to obtain and carry out experiments to obtain data, they should include additional measurements such as correlation experiments and others (HMBC, HSQC, COSY) , which would give better support to the proposed structure.

All experiments should be mentioned in this section.

In the results section, there is mention of NOESY spectra, however in this section it is not mentioned?'.

4-    Please include as supplementary material all NMR spectra in the best possible quality.

5-    Section 4.6. Determination of antioxidant capacity of 2,2-diphenyl-1-picrylhydrazyl (DPPH) assay

“The results were expressed as the relative concentration required to reduce the DPPH concentration by 50% (EC50). Ascorbic acid was used as positive control of antioxidant activity in both assays.”

However, in the results section, the data is presented as scavinging  activity in percentage (Figure 5), there is also no mention of EC50 in the discussion section.

This should be reviewed and improved

The manuscript, after the suggested changes, should be reviewed again for its potential acceptance.

Moderate editing of English language

Author Response

Manuscript ID: molecules-2386105

A New Lignan from Annona squamosa L. (Annonaceae) demonstrate vasorelaxant effects in vitro

Dear Section Managing Editor

Abigail Zhao,

Thank you for the opportunity to keep the review process in order to publish in Molecules, “Antioxidant Activity of Natural Products” Topic.

It is a great pleasure for our research group to clarify the points highlighted by the reviewers. The manuscript was adjusted in order to attend the reviewer’s comments. For the best comprehension of all modifications made, please find below the answers for each item pointed out.

Reviewer 1:

The information included for structure 1 could be improved and supported with additional experiments considering the availability of powerful NMR equipment.

Thank you for the suggestion. A better discussion on the structural elucidation esquamosan can be found in lines 90-120.

1-    Please, assign a number to Figure 1. Structure of squamosan. , which will allow a better reading of Table 1.

 The proposed structure for esquamosan is  currently presented together com the structure of the other isolated compounds, without belonging to a Figure (page 4)

2-    Please, assign a number to the carbons in Figure 1, for a better reading of the 13C data that you mention in the section 4.4. Spectral Data 13 C.

 The numbers of the positions were assigned within the structure of compound 1. The chemical shifts of the carbons were placed in table 1 and removed from section 4.4

3-    Section  4.1. General Information

Considering that the authors have a robust NMR team to obtain and carry out experiments to obtain data, they should include additional measurements such as correlation experiments and others (HMBC, HSQC, COSY) , which would give better support to the proposed structure.

 The NMR experiments were performed again in a Bruker 600 spectrometer using deuterated chloroform in order to make a better comparison with the spectroscopic data reported in the literature for membrine and epimembrine. All 2D NMR spectra were included in the supplementary material and mentioned in the section.

All experiments should be mentioned in this section.

done

In the results section, there is mention of NOESY spectra, however in this section it is not mentioned?'.

 NOESY experiments  were included in section 4.1

4-    Please include as supplementary material all NMR spectra in the best possible quality.

All 1D and 2D NMR spectra as well as the HRMS spectrum were included in the Supplementary Materials section.

5-    Section 4.6. Determination of antioxidant capacity of 2,2-diphenyl-1-picrylhydrazyl (DPPH) assay

 “The results were expressed as the relative concentration required to reduce the DPPH concentration by 50% (EC50). Ascorbic acid was used as positive control of antioxidant activity in both assays.”

However, in the results section, the data is presented as scavinging  activity in percentage (Figure 5), there is also no mention of EC50 in the discussion section.

The following paragraph was included in section 4.6:

 EC50 values in the DPPH assay were 56.8 ± 1.3 and 5.39 ± 0.1 μg/mL (159.4 and 30.6 μM) and for escuamosan and ascorbic acid, respectively. The FRAP     activity results were 32.6 ± 6.5 and 8.9 ± 0.3 μg/mL equivalents to 100 µM of Fe2+ for escuamosan and ascorbic acid, respectively.

The manuscript, after the suggested changes, should be reviewed again for its potential acceptance.

Reviewer 2 Report

Your manuscript could be publish just some punctuation corrections should be done. 

Author Response

Manuscript ID: molecules-2386105

A New Lignan from Annona squamosa L. (Annonaceae) demonstrate vasorelaxant effects in vitro

Dear Section Managing Editor

Abigail Zhao,

Thank you for the opportunity to keep the review process in order to publish in Molecules, “Antioxidant Activity of Natural Products” Topic.

It is a great pleasure for our research group to clarify the points highlighted by the reviewers. The manuscript was adjusted in order to attend to the reviewer’s comments. For the best comprehension of all modifications made, please find below the answers for each item pointed out.

Reviewer 2:

Thank a lot for your suggestions, the following changes were made:

1-The name A.squamosa was changed by Annona squamosa, line 59 and 312.

2- catechin (1) was changed by catechin (7), line 83

3-The chemical shift values were corrected in the text and in Table 1. In order to compare the spectroscopic of compound 1 with the NMR data of membrine and epimembrine, the 1D and 2D NMR experiments were performed again in a Bruker 600 spectrometer, using deuterated chloroform as a solvent.  

4- the word cis was changed by axial.

Reviewer 3 Report

Di Giulio et al. isolated a new compound, escuamosan, from the leaves of Annona squamosa and elucidated its structure based on mass spectrometry and NMR measurements. In a second part the authors investigated the vascular effects in vitro by using rat thoracic aorta rings and finally also in vitro the antioxidant and reducing effects of this compound. Altogether 24 references were considered in this manuscript.

The authors have selected an interesting and promising topic covering possible future medicinal uses of the leaves of Annona squamosa and add to previous research on this herbal material.  

HOWEVER, there are several issues which should be reconsidered by the authors:

General:

1.    The name of the new compound esquamosan should be written in the same way throughout the whole manuscript. Please check again!

2.    All botanical plant names in the manuscript and also in the list of references need to be written in italics (see e.g. line 59, 61 and 261, and also lines 349-403).

3.    Many abbreviations are used, like „HG“, „Ach“, „PE“. A list of abbreviations should be added at the end of the manuscript.

Specific aspects:

4.     Line 44: it should be „Annona squamosa L.“ (i.e. the full botanical name including the author´s name, Linné, needs to be given).

5.     Line 45: Please add the traditional medicine of …, i.e. please specify the country an/or region.

6.     Lines 59-84: The isolation procedure is complex. Therefore I propose to add a chart on this process. Otherwise the readers might have difficulties to follow the separation and isolation processes.

7.    Line 80: Where does fraction IV comes from? You only mentioned fractions I-III in line79/80, but not a fraction IV!

8.    Lines 71-84: More details on the structure elucidation on the kauren derivatives and the flavonoids should be presented.

9.    Line 75: Ma et al., 2017 should be reference 20. Please add the number of the reference in the text instead of „Ma et al., 2017“!

10. Lines 85-86: The structure elucidation is incomplete. It is not sufficient to mention only a [M+Na]+ peak, but also prominent fragments and/or cleavage fragments should be presented with their intensities as these fragments also contribute to a proper structure elucidation.

11. Lines 88-89: The values for the shifts are not congruent with the data given in Table 1. Please correct the data!

12. Lines 89-90: Please explain in more detail the substitution patterns of the aromatic ring systems, i.e. how was the presence of a 4-methoxy phenyl and a 3,4-dimethoxy phenyl group confirmed? Which is the important feature four your observation?

13. Lines 90-93: Please indicate where differences in the relative orientation can be seen, i.e. which is the important feature in the data for this? Otherwise your sentence and your conclusion is difficult to understand for the audience.

14. Line 107 / Figure 1: The numbers of the C-atoms in the formula should be added. Otherwise the readership cannot follow the data given in Table 1.

15. Line 110: „Epimembrine“ and „Membrine“

16. Line 110-111 / Table 1: Please add whether you performed the NMR measurements on epimembrine and membrine on your own (i.e. parallel to compound 1) or whether you used literature data? I assume that you used data from reference 16 (Estrada-Reyes et al., 2002), but this urgently needs to be added in a scientific sound manuscript!

17. Line 133-135 / Figure 2: The legends for figures 2C and 2D are incomplete. Please add explanations in the legend. Please also add the number of replicates!

18. Lines 112-131: Did you use a reference compound? This would be very helpful in order to evaluate the effect of escuamosan, i.e. in order to compare the results and the size of the effects observed compared to standard compounds.

19.  Line 150 and other lines. „Bay K8644“. Please check the spelling in the manuscript!

20. Line 155: the concentrations „1, 10 and 30 µM“ are not the same as given in the legend for Figure 3 in line 177 („1, 5 or 10 µM“). Please check again and correct and/or explain!

21. Lines 152-172: The description of the results are extremely diffult to understand, as the curves n Figure 3 are not labelled. The reader can not see, which curve belongs to which concentration tested. Thus, please, add the concentrations in each of the three diagrams! The number of replicates nees to be added.

22. Line 208: Please check the sentence again (which lignan from Brazilian plants?)!

23. Lines 205-236: Please check whether comparable furofuran lignans have previously been tested for vasorelaxant effects and add this in your discussion, i.e. comparison with your results!

24. Line 224: Please complete the sentence „majority of reported“ (Do you mean majority of authors?)

25. Line 262: Please add the place of employment and/or the kind of employment (e.g. University of …, Dept of Botany or Hort. Garden of …)

26. Line 272: „(40 mg/kg body weight, intraperitoneal) …“

To sum up, this is an interesting study which will be of great interest for the international readership of the journal. However, the issues mentioned, especially regarding a proper and convincing structure elucidation and also the presentation of the diagrams and their legends should thoroughly be addressed by the authors.

The whole manuscript should be thoroughly checked for spelling and grammar mistakes, including the abstract!

Author Response

Reviewer 3:

Di Giulio et al. isolated a new compound, escuamosan, from the leaves of Annona squamosa and elucidated its structure based on mass spectrometry and NMR measurements. In a second part the authors investigated the vascular effects in vitro by using rat thoracic aorta rings and finally also in vitro the antioxidant and reducing effects of this compound. Altogether 24 references were considered in this manuscript.

The authors have selected an interesting and promising topic covering possible future medicinal uses of the leaves of Annona squamosa and add to previous research on this herbal material. 

HOWEVER, there are several issues which should be reconsidered by the authors:

General:

  1. The name of the new compound esquamosan should be written in the same way throughout the whole manuscript. Please check again!

Done. The text  was  revised and changed when pertinent to  esquamosan

  1. All botanical plant names in the manuscript and also in the list of references need to be written in italics (see e.g. line 59, 61 and 261, and also lines 349-403).

All botanical plant names were reviewed and written in italics.

  1. Many abbreviations are used, like „HG“, „Ach“, „PE“. A list of abbreviations should be added at the end of the manuscript.

A list of abbreviation was added at the end of the article

Specific aspects:

  1. Line 44: it should be „Annona squamosa L.“ (i.e. the full botanical name including the author´s name, Linné, needs to be given).

Done.

A more detailed description of the isolation process was written in section 2.1. Table 1 was modified and a summary scheme of the separation process was added.

  1. Line 45: Please add the traditional medicine of …, i.e. please specify the country an/or region.

Done. The text was modified to “]. Annona squamosa L. is a tree that has multiple pharmacological applications in the Indian traditional medicine, including its popular use for the treatment of cardiometabolic disease, which has been validated in experimental models”(44-47)

  1. Lines 59-84: The isolation procedure is complex. Therefore I propose to add a chart on this process. Otherwise the readers might have difficulties to follow the separation and isolation processes.

Done. A figure containing the main steps of extraction and separation was  included in Material and Methods section 4.4

  1. Line 80: Where does fraction IV comes from? You only mentioned fractions I-III in line79/80, but not a fraction IV!

    We apologize for this error, from the separation of the AF fraction only three subfractions were obtained and were called I, II and III. The phrase fraction IV was changed to fraction III.

  1. Lines 71-84: More details on the structure elucidation on the kauren derivatives and the flavonoids should be presented.

     Kauren-type diterpenes isolated from MWIF-II and flavonoids isolated from AF-III were previously reported in Anona squamosa by Ma et al. 2017 and Zhu et al. 2020. We respectfully think that it is not necessary to describe the structural elucidation of these compounds because they do not represent a novelty for the present work. The MWIF-II fraction showed a slight vasorelaxing effect and kaurenoic acid was identified in it, which had shown a vasorelaxing effect in previous works. In our opinion, the MWIF-II subfraction does not demonstrate the most important vasorelaxing effect and therefore we assume that the slight effect must be due to the presence of kaurenoic acid. The values of the weights obtained from all the compounds were added. The citation to the work of Zhu et al. was added to the reference list.

  1. Line 75: Ma et al., 2017 should be reference 20. Please add the number of the reference in the text instead of „Ma et al., 2017“!

I apologize for the mistake. All Citation numbers were  checked and changed if pertinent

  1. Lines 85-86: The structure elucidation is incomplete. It is not sufficient to mention only a [M+Na]+ peak, but also prominent fragments and/or cleavage fragments should be presented with their intensities as these fragments also contribute to a proper structure elucidation.

The most prominent molecular ion and fragments with their relative intensities were referenced in section 4.5. In the supplementary material section, the structures were added to the fragments in the mass spectrum.

  1. Lines 88-89: The values for the shifts are not congruent with the data given in Table 1. Please correct the data!

 The data were corrected

  1. Lines 89-90: Please explain in more detail the substitution patterns of the aromatic ring systems, i.e. how was the presence of a 4-methoxy phenyl and a 3,4-dimethoxy phenyl group confirmed? Which is the important feature four your observation?

The explanation of the structural elucidation of compound 1 was redrafted.

  1. Lines 90-93: Please indicate where differences in the relative orientation can be seen, i.e. which is the important feature in the data for this? Otherwise your sentence and your conclusion is difficult to understand for the audience.

A new explanation about the structural elucidation of compound 1 was made. Basically, we explain that the difference among these compounds is inferred primarily by the differences in the coupling constants and is reaffirmed by the interactions that are observed in the NOESY spectrum.

  1. Line 107 / Figure 1: The numbers of the C-atoms in the formula should be added. Otherwise the readership cannot follow the data given in Table 1.

Done

  1. Line 110: „Epimembrine“ and „Membrine“

The names Epimembrine and Membrine were corrected in the text.

  1. Line 110-111 / Table 1: Please add whether you performed the NMR measurements on epimembrine and membrine on your own (i.e. parallel to compound 1) or whether you used literature data? I assume that you used data from reference 16 (Estrada-Reyes et al., 2002), but this urgently needs to be added in a scientific sound manuscript!

    We thought it might be interesting to compare the spectroscopic data of the already known compounds with compound 1. However, in the new explanation for the structural elucidation of esquamosan, we focus on the comparison of the coupling constants of the protons at the 7-positions , 8 and 9, with the isomers already reported and the information obtained with the NOESY experiment. Table 1 was modified, the values of the displacements of the C13 spectrum were included and the data reported by Estrada et al 2017 were eliminated.

  1. Line 133-135 / Figure 2: The legends for figures 2C and 2D are incomplete. Please add explanations in the legend. Please also add the number of replicates!

In this revised version of the manuscript, figure 2 is now figure 1, the explanation of the legend in figure 1 was improved and the number of replicates in all figures was added.

  1. Lines 112-131: Did you use a reference compound? This would be very helpful in order to evaluate the effect of escuamosan, i.e. in order to compare the results and the size of the effects observed compared to standard compounds.

   We did not use a positive control in order to compare  the vasorelaxing effect of esquamosan. However, we can make two comparisons regarding other compounds. First in intact aortic rings escuamosan has comparable potency with 1 µM vs 0.3 µM acetylcholine, which was used as the positive control in assays of aortic rings exposed to high glucose concentrations, although the mechanisms of action are different, Ach acts through M3 receptors in the vascular endothelium and induces the formation of NO, which finally acts as a vasorelaxant. On the other hand, when comparing the IC50 values of esquamosan with that of yangambin, another lignan furofuran with a vasorelaxing effect reported in the literature, squamosan is almost 100 times more potent than yangambin, 1 µM vs 100 µM. Yangambin has three additional methoxyl groups  and the configuration of the aromatic rings is the opposite. We might suggest that squamosan is more potent than yangambin because of these differences. There is a similarity in the vasorelaxing effect of both compounds, according to the reported data, Yangambin exerts its effect through the blockade of calcium channels, in that work they used Nifedipine as a positive control (IC50 = 10 nM). Although the effect of Yangambin is reached at much higher concentrations, in that work the blockade of voltage-gated calcium channels was clearly demonstrated using the agonist (S)-(−)-BAY-K-864420.

  1. Line 150 and other lines. „Bay K8644“. Please check the spelling in the manuscript!

The name of the reagent BAY K8644 was changed in the text to (S)-(−)-BAY-K-864420.

  1. Line 155: the concentrations „1, 10 and 30 µM“ are not the same as given in the legend for Figure 3 in line 177 („1, 5 or 10 µM“). Please check again and correct and/or explain!

     There was a mistake when writing the value of the concentrations used in Figure 3. Note that in the other figures it is described that the fixed value used for the experiments is 1 µM, which corresponds to the reported IC50 value.

  1. Lines 152-172: The description of the results are extremely diffult to understand, as the curves n Figure 3 are not labelled. The reader can not see, which curve belongs to which concentration tested. Thus, please, add the concentrations in each of the three diagrams! The number of replicates nees to be added.

The name of figure 3 was changed by figure2 , the description of figure 2 was improved and the number of replicates was added.

  1. Line 208: Please check the sentence again (which lignan from Brazilian plants?)!

This phrase was deleted and was in the original text by mistake. In general, lignans are a group of compounds that have shown a wide variety of biological effects. The reference talks about lignans isolated from plants from Brazil, particularly Yangambin, but regardless of the country of origin, this type of compound has demonstrated various biological activities, including vasorelaxing.

  1. Lines 205-236: Please check whether comparable furofuran lignans have previously been tested for vasorelaxant effects and add this in your discussion, i.e. comparison with your results!

 The discussion seccion is now at the line 232, to the best of our knowledge, lignans such as eudesmin, saucerneol , saucerneol D , saucerneol E , manassantin A, Cudebin, gomision J , have shown a vasorelaxing effect dependent on vascular endothelium in experiments performed on rat aorta rings. Only Phyllanthin, Hypophyllanthin, Yagambin and now squamosan have demonstrated a vasorelaxing effect independent of the vascular endothelium and mediated by blocking the movement of calcium within smooth muscle. It would be interesting for future molecular modeling studies to find out which are the common structural characteristics that give these compounds such vasorelaxing properties. Additionally, we compared the antioxidant effect of esquamosan with that of other lignans.

  1. Line 224: Please complete the sentence „majority of reported“ (Do you mean majority of authors?)

The sentence was changed to:” Most of the mechanisms reported in the literature underlying hyperglycemia-induced diabetic vascular damage focus on five main mechanisms”…line 248.

  1. Line 262: Please add the place of employment and/or the kind of employment (e.g. University of …, Dept of Botany or Hort. Garden of …)

Done. The text was modified to “Anonna squamosa leaves were collected in September 2018 at the National Park Henri Pittier (Edo Argua, Venezuela). The species was identified by Dr. Alfonso Cardozo (Department of Botany, Venezuelan Central University). Central de Venezuela. A voucher specimen was deposited in the herbarium at the Agronomy Faculty of the Venezuelan Central University (voucher number: AC21992).

  1. Line 272: „(40 mg/kg body weight, intraperitoneal) …“

done

To sum up, this is an interesting study which will be of great interest for the international readership of the journal. However, the issues mentioned, especially regarding a proper and convincing structure elucidation and also the presentation of the diagrams and their legends should thoroughly be addressed by the authors.

Comments on the Quality of English Language

The whole manuscript should be thoroughly checked for spelling and grammar mistakes, including the abstract!

Thank you for the valuable suggestions.

Round 2

Reviewer 1 Report

Suggestions or changes have been considered by the authors. The manuscript should be accepted for publication.

Moderate editing of English language

Author Response

Thank a lot for your valuable suggestions.

Reviewer 3 Report

Dear authors,

thank you very much for your cover letter and for a new version of the manuscript based on the reviewers´ comments. In my point of view the manuscript has extremely been improved. My comments (Reviewer 3) were fully considered and all changes were very thoroughly performed. Additional explanations were added. Therefore I recommend the acceptance  of the paper.

Only some last very minor aspects should be considered prior to acceptance:

A.    Line 59: Please use the correct plant name!

B.    Line 122: This is an excellent idea to add the structural formulas of the compounds isolated within this study. It is up to the authors to decide whether this should be named figure 1, but this is only a proposal!

C.    Line 314 and line 483: Please write the plant name in italics!

D.   Line 322: Please write 399.1772!

E.    Line 434 species

In conclusion this is a most interesting and excellent study combining phytochemistry and pharmacology. This study will be of great interest to the readers of the journal!

Kind regards, reviewer 3

Author Response

  1. Line 59: Please use the correct plant name!

Done.

  1. Line 122: This is an excellent idea to add the structural formulas of the compounds isolated within this study. It is up to the authors to decide whether this should be named figure 1, but this is only a proposal!

Thank you for your kind comment, we prefer to keep the structural formulas of the isolated compounds

  1. Line 314 and line 483: Please write the plant name in italics!

Done

  1. Line 322: Please write 399.1772!

Done, we believe the reviewer is referencing the signal at 395.1445

  1. Line 434 species

Done

Thanks a lot for your kind comments.
